# An Efficient Pareto Optimal Resource Allocation Scheme in Cognitive Radio-Based Internet of Things Networks

**DOI:** 10.3390/s22020451

**Published:** 2022-01-07

**Authors:** Shahzad Latif, Suhail Akraam, Tehmina Karamat, Muhammad Attique Khan, Chadi Altrjman, Senghour Mey, Yunyoung Nam

**Affiliations:** 1Shaheed Zulfikar Ali Bhutto Institute of Science and Technology (SZABIST), Islamabad 75600, Pakistan; dr.shahzad@szabist-isb.edu.pk; 2BARANI Institute of Sciences Burewala, JV of PMAS ARID Agriculture University Rawalpindi, Rawalpindi 46000, Pakistan; sohail.a@baraniinstitute.edu.pk; 3Department of Software Engineering, Foundation University Islamabad, Rawalpindi 46000, Pakistan; tehminakaramat@fui.edu.pk; 4Department of Computer Science, HITEC University Taxila, Taxila 47080, Pakistan; attique.khan@hitecuni.edu.pk; 5Artificial Intelligence Engineering Department, AI and Robotics Institute, Near East University, Mersin 99138, Turkey; cmfaltrj@uwaterloo.ca; 6Department of ICT Convergence, Soonchunhyang University, Asan 31538, Korea; meysenghour2888@gmail.com; 7Department of Computer Science and Engineering, Soonchunhyang University, Asan 31538, Korea

**Keywords:** pareto optimality, energy efficiency, spectral efficiency, resource allocation, CR-IoT networks

## Abstract

The high data rates detail that internet-connected devices have been increasing exponentially. Cognitive radio (CR) is an auspicious technology used to address the resource shortage issue in wireless IoT networks. Resource optimization is considered a non-convex and nondeterministic polynomial (NP) complete problem within CR-based Internet of Things (IoT) networks (CR-IoT). Moreover, the combined optimization of conflicting objectives is a challenging issue in CR-IoT networks. In this paper, energy efficiency (EE) and spectral efficiency (SE) are considered as conflicting optimization objectives. This research work proposed a hybrid tabu search-based stimulated algorithm (HTSA) in order to achieve Pareto optimality between EE and SE. In addition, the fuzzy-based decision is employed to achieve better Pareto optimality. The performance of the proposed HTSA approach is analyzed using different resource allocation parameters and validated through simulation results.

## 1. Introduction

In the future, it has been forecasted that billions of devices will be connected through IoT technology. Data rates are expected to raise many fold in coming years across the globe. Future networks will come to a standstill if further capacity is not created [1]. The Internet of Things (IoT) is a framework that is being used to connect wireless devices. There are numerous IoT applications in different fields of life, such as medical, agriculture, education, transportation, etc., [2]. The IoT systems can save a lot of human resources and will generate businesses. Due to the minimum energy resources of IoT systems, efficient energy and spectrum allocation are the most important factors in IoT communication [3,4]. On the other hand, CR communication is a promising technology to deal with spectrum allocation problems. CR is an intelligent device that scans the spectrum around its vicinity and searches vacant spectrum locations both in time and frequency which are not being utilized by licensed users also known as primary users (Pus) [5]. The cognitive radio network (CRN) connects cognitive radios that share the vacant spectrum among unlicensed users also known as secondary users (Sus), provided that they do not create interference with Pus [6]. Moreover, CR communication is considered green communication because it tries to maximize the spectrum efficiency with minimum power interference with Pus [7].

Therefore, the combination of CR technology with IoT systems can improve EE and spectral efficiency which will be a great boom in IoT industry. The CR-IoT is a proposed technology for fifth generation (5G) networks. Due to this potential performance improvement, CR-based IoT (CR-IoT) has attained the attention of researchers. EE is defined as calculating the throughput concerning total power consumed [8,9]. In [10], EE for joint base station and beamforming in the multicell scenario is investigated. In [11,12], energy and spectrum efficiency in 5G mobile multiple input multiple output (MIMO) networks are examined. The authors in [13], proposed a non-cooperative energy-efficient game for distributed CRN over interference channels. A stochastic Stackelberg game is studied for balancing network delays and power allocation in energy harvesting CRN. A power allocation-based noncooperative game is proposed for CRNs and IoT [14]. In [15], a mesh adaptive search algorithm is examined for the device to device-assisted CRN. In [16], a gradient adaption-based optimization is proposed for power allocation and EE in CRNs. The gradient methods are robust, but sometimes they fail to achieve global optimization. 

To reduce the computational complexity of optimization approaches, heuristic algorithms are gaining the attention of the researchers. Heuristic approaches are easy to implement and flexible for NP complete problems [17]. Joint optimization of EE and spectral efficiency are considered as non-convex and NP hard problem in CR-IoT systems [18]. In [19], IoT resource management using non orthogonal multiple access (NOMA) scheme is used for CR-IoT in smart cities and mixed integer linear programming is proposed for energy harvesting. The mixed integer nonlinear programming (MINLP)-based approach is proposed to optimize the EE and spectral efficiency trade off in CR-IoT. 

Meta heuristic approaches are also widely used to optimize resource allocations in CR-IoT systems [20,21,22,23]. Meta heuristic approaches are good to deal with the multi objective optimization problems and flexible to deal with multiple constraints. However, heuristic algorithms have convergence issues, and computational complexity has increased with the increasing size of the population [24]. 

The hybrid meta heuristic algorithm can enhance the performance of optimization problems by combing the exploration and exploitation features of different algorithms [25]. Hybrid meta heuristic algorithms have not been studied much in the literature. EE and spectral efficiency are considered as conflicting objectives [26].

To the best of our knowledge, hybrid meta heuristic-based approaches are not used much for conflicting objectives in CR-based IoT systems. Meta heuristic approaches are easy to implement and have low computational complexity. 

The main contributions of this paper are as follows:Propose an optimization algorithm for conflicting optimization objectives.Analyze the performance of proposed model with different performance metrics and compare with other optimization approaches.

The rest of the paper is organized as follows: Section 2 describes the system model of CR-IoT. Section 3 provides the proposed algorithm based on HTSA. The performance of proposed methodology is analyzed in Section 4. Finally, the paper is concluded in Section 5.

## 2. System Model

Figure 1 represents describes the system model of CR-IoT. The IoT nodes are considered as Sus, which can utilize the spectrum resources opportunistically and cellular users are considered as Pus. There are *N* number of IoT nodes are considered. There is a link between each pair of nodes of transmitter and receiver. Let *M* denote a set of free channels. It is assumed that numbers of Sus are greater than number of available channels, i.e., *N > M*. The availability of free channels for supporting Sus data transmission depends upon the Pus activity and also Sus activity. Let *l(I,j)* denotes the link between *i^th* and *j^th* node of CR-IoT network. A spectrum channel and certain amount of power are allocated to each link *l(i,j)* for data transmission. *F* is the list of data flows in the network. In Figure 1, three are three data flows *f1*, *f2* and *f3*, respectively.

Let P(i,j) represent the power consumed by link l(i,j) under the following constraint
$$P_{min}\leq P\left(i,j\right)\leq P_{max}$$
where $P_{min}$ and $P_{max}$ denote the minimum and maximum power consumption thresholds, respectively. In a wireless network, data rates vary over different links depend on many factors such as signal to interference plus noise ratio (SINR) and fading. The SINR can be expressed as [27].
(1)$$SINR_{\left(i,j\right)}\left(t\right)=\frac{g_{i,j}P_{i}\left(t\right)}{N+\sum_{\left(a,b\right)\in L\left(a,b\right)\neq \left(i,j\right)}g_{aj}p_{a}}$$
where $g_{i,j}$ denote channel gain between transmitter i and receiver j, and represented by $\frac{k}{d_{i,j}^{\alpha }}$. $d_{i,j}$ correspond to the distance between nodes i and j, α is the path loss coefficient. According to (1), the SINR decreases with the distance between to nodes. $P_{i}\left(t\right)$ denotes the transmission power of ith transmitter at time t, $N\left(o,\sigma^{2}\right)$ is additive white Gaussian noise with zero mean and variance $\sigma^{2}.$ L is the set of links sharing the spectrum channel m. The link (i,j) can transmit dataflow f on the channel m if the following constrained satisfied.
(2)$$SINR_{\left(i,j\right)\;}\left(t\right)\geq \gamma$$
where γ denote the threshold value of SINR to maintain minimum quality of service (QoS). In CRN, each node can either send or receive data at given time instant t. The throughput of link l(i,j) at given time t can be represented as
(3)$$R_{\left(i,j\right)\;}\left(t\right)=x_{i,j}Wlog_{2}\left(1+SINR_{\left(i,j\right)\;}\left(t\right)\right)$$
where W represents the bandwidth of the spectrum channel and $x_{i,j}$ denotes binary decision variable. If $x_{i,j}=1$, ith transmitter node and jth receiver node can access the vacant channel and vice versa in CR-IoT network. In (3), the throughput can be maximized by increasing SINR, which means power consumptions will be increased. Hence, there is a tradeoff in maximizing throughout and minimizing power consumption. To maintain the minimum data rate requirement for each SE node, the following fairness criteria are considered:(4)$$R_{fair\left(i,j\right)\;}\left(t\right)=\max\left\{\min\left({\frac{1}{L}}_{i,j}\right)\times M_{i,j}\;\right\}\;\forall \;i,j\in N$$

The above expression (4) represents the maximized throughput of those SU nodes which are using maximum data links and minimum number of spectrum channels.

The EE can be described as the ratio of the network throughput to power consumption [27].
(5)$$F_{EE}\left(t\right)=\frac{Total\;network\;Throughput}{Total\;Power\;Consumtion}=\frac{\sum_{f\in F}\sum_{\left(i,j\right)\in L^{f}}R_{\left(i,j\right)}\left(t\right)}{\sum_{f\in F}\sum_{\left(i,j\right)\in L^{f}}P_{\left(i,j\right)}\left(t\right)}$$
where $L^{f}$ is the list of links that constitutes the data flow f. The data rate may vary enormously between links in each data flow. Spectrum utility $F_{Spec}\left(t\right)$ can be defined as the ratio of number of link nodes to number available spectrum channels M(t) at time t.
(6)$$F_{Spec}\left(t\right)=\frac{\left|L\right|}{\left|M\left(t\right)\right|}$$
where M(t) is total number of available channels to N Sus at time t.

## 3. Proposed Hybrid Simulated-Tabu-Based Resource Optimization Algorithm

The EE and spectral efficiency are the desired objectives which we want to maximize. However, these objectives are conflicting with each other. If we want to maximize the throughput then EE cannot be maximized due to more power consumption in increasing throughput. Following are the two objectives which want to maximizes

In multi-objective optimization problems (MOOP), a single optimal solution cannot be defined; rather, a set of solutions is considered, known as the pareto optimal front. 

The non-dominated solutions (NS) that fulfilled the above criteria constitute a pareto optimal front. It is very difficult to find unknown pareto optimal fronts in MOOP. Hence, the set of NS provide an approximation to pareto optimal front. Such kind of problems generally requires high computational complexity. Evolutionary computing algorithms are good to deal with pareto optimality in multi objective scenarios [28]. This research work proposes a hybrid multi-objective optimization algorithm called hybrid tabu search-based stimulated annealing (HTSA) combing the features of tabu search and stimulating annealing.

### 3.1. Simulated Annealing (SA)

SA is mostly used to find the global optimum. The SA takes its inspiration from annealing process of solids in which a solid shape is formed by heating a solid. In the annealing process, a high temperature solid is steadily cooled down so that atoms reach the stable or equilibrium state [29]. SA used this concept to find the optimal solution using a stochastic search. SA searches the neighboring states or solutions and accepts it if its probability is above a certain threshold. The probability function depends on the temperature (T) parameter. In SA, the solid energy state is considered as a viable solution, and improvement in the energy state is considered to be upgrading in the objective function. Finally, if there is no further change in the energy state then the final energy state is considered as the optimal solution.

### 3.2. Tabu Search (TS)

TS was proposed by Glover [30] and provides a near-optimal solution. TS moves from a weak solution to better solution by moving or searching the neighbor’s space. Principally, a complete list of neighbors should be explored to find the best neighbor solution. However, a complete neighborhood search increases the computational complexity drastically. TS maintains a tabu list (TL), which stores a subset of neighbors.

TL helps to explore new solution space while avoiding being stuck in cycles. TL contains the list of moves that produce good solutions in previous searches. If the cost of the move is better than the previous move, then the move can be carried out, otherwise not. TL contains a queue of moves. When the queue is filled, old moves are removed, and new moves are added to the list. The choice of move depends on the lowest cost associated with the move. 

### 3.3. Hybrid Tabu Simulated Algorithms (HTSA)

Due to the stochastic nature of SA, the algorithm explores more search space; therefore, it has difficulty remaining at local minima. At the start, weak solutions are accepted due to the high temperature. However, with the passage of time, the solution space is improved due to temperature reduction. In contrast, TS is not stochastic in nature, exploiting the past search experience and maintaining a candidate list to avoid cycling. Thus, HTSA can provide efficient solutions. The combination of exploration capabilities of SA and exploitation capabilities of TS can achieve better performance. This search process is also explained in Figure 2. The HTSA is divided in two phases: the first phase contains TS and second phase consists of SA. In first phase, TS performs movements, and in the second stage, stimulated annealing performs temperature-based operations to find the optimal value. $O_{1}$ and $O_{2}$ are objectives defined in Equations (3) and (4), respectively. The new solution $q^{*}$ is dominated by another solution q if the following conditions satisfied.
(7)O1(q*)≥O2(q) ^O2(q*)≥O1(q)
where $O_{1}\left(q^{*}\right)$ and $O_{2}\left(q\right)$ are the values of objective functions for solutions $q^{*}$ and q, respectively, and vice versa. The dominated solutions are accepted with probability one. In contrast, if $q^{*}$ does not dominate q, then $q^{*}$ is accepted with the following probability.
(8)$$\overset{2}{\underset{k=1}{\prod }}\min\left[1;e^{\left\{\frac{O_{1}\left(q\right)-O_{2}\left(q^{*}\right)}{t}\right\}}\right]$$

If both q and $q^{*}$ not dominates each other, then the probability of accepting a new solution is
(9)$$1-\frac{card\left(D\left(q^{*}\right)\right)}{\left|P\right|}$$
where |P| represent the population size of solution space. $D\left(q^{*}\right)$ is crowding distance which is used to compare the solutions belonging to the same non dominating front. The new solution is accepted with high probability if it is located near to other feasible solutions and vice versa. The crowding distance is calculated as [31].
(10)$$D\left(q^{*}\right)=\overset{2}{\underset{k=1}{\sum }}\frac{O_{k}\left(q^{*}+1\right)-O_{k}\left(q^{*}-1\right)}{O_{k}^{max}-O_{k}^{min}}$$

The SUs nodes are considered as the initial population and generated randomly. The values of initial temperature Ti, cooling temperature rate Tc, and stop criteria Tstop are assigned. The temperature values are decreased with increasing iterations *k* and are defined as
(11)T=1−k+1kmax
where kmax denotes totoal number of iterations. The TL is initially empty and updated by removing those movements which are forbidden in previous iterations. After this, the tabu search TS starts, in which those non tabu movements are selected, which dominates the previous solution. This process is continued until better movements are obtained. When no further improvement found, this loop stops. Next, the archiving loop is applied. In this case, a new solution is randomly generated. This new solution is accepted or rejected according to the crowding criteria discussed in Equations (9) and (10), respectively. After that, the population is updated by applying a selection process, and the external archive NS is considered. When the loop is finished, the set of non-dominated solutions NS is obtained. The procedure of algorithm is also explained in Algorithm 1. The values used in the HTSA are indicated in Table 1.
**Algorithm 1: HTSA Procedure.**Initialize: Population size N, Ti,Tc, Tstop,q, TL(q)Generate the initial solution randomly and RepeatFor (∀ q ∈Q) do Calculate the values the objective functions defined in Equations (3)–(5) and then obtain no dominating fronts based on Equations (7)–(9)Update TL(q)If (∃ move (q,q*)∉TL: q*≺q) thenAssign q=q*; stop TS;TL=TL∪move (q,q*);else If (∃ q*∈NS:q ≺(domnates) q*) thenq*=q;NS=NS−Q;else if NS=NS ∪q ;T=T*TcReturn NSUntil (*T* > Tstop)

### 3.4. Fuzzy-Based Final Decision Making

The final pareto optimal solution obtained from HTSA may have imprecise judgment due to the heuristic nature of the algorithm. Hence, fuzzy-based decision making is considered for the best compromised solutions. The fuzziness is defined by the following membership function [32]:(12)$$\mu_{i}^{q}=\{\begin{array}{c} 1\;if\;O_{i}\geq O_{i}^{max}\\ \frac{O_{i}^{max}-O_{i}}{O_{i}^{max}-O_{i}^{min}}\;if\;O_{i}^{min}<O_{i}<O_{i}^{max}\\ 0\;if\;O_{i}<O_{i}^{min} \end{array}$$
where $\mu_{i}^{q}$ is the membership value of the q non dominated solution for the ith objective. The sum of $\mu_{i}$ for all the objectives of the q solution determines the quality of the solution, and is defined as
(13)$$\mu^{q}=\frac{\sum_{i=1}^{2}\mu_{i}^{q}}{\sum_{q=1}^{Q}\sum_{i=1}^{2}\mu_{i}^{q}}$$
where i=2 represents the proposed two objectives. The highest membership value of $\mu^{q}$ can be accepted by the decision makers.

## 4. Results and Discussion

In this section, the proposed algorithm performance is evaluated through simulations in different scenarios. The system parameters are represented in Table 2. The CR-IoT networks are deployed over a 1000 m × 1000 m area and consist of static IoT nodes that constitute different data flows and are connected through links, as shown in Figure 3. To evaluate the performance of proposed algorithm, simulations are conducted, and results are compared with standard SA and TS. 

Figure 3 showed the pareto optimality between energy efficiency and spectrum efficiency for various SINR. Figure 4a,b illustrate pareto fronts between the energy efficiency and spectrum efficiency for SINR = 4 db and 6 dB, respectively. For low SINR thresholds, less network throughput is achievable due to reduced energy efficiency. Link capacity is also reduced because fewer channels are assigned to varying SUs nodes. Figure 4c,d indicates the pareto fronts for higher SINR of 8 dB and 10 dB, respectively. Figure 4c,d results showed that pareto front value are increasing linearly with increasing SINR. Figure 3 overall results showed that proposed HTSA performed better than conventional SA and TS. The best solutions are obtained in HTSA due to combining the exploration features of SA and exploitation feature of TS. Exploration corresponds to searching efficiency and exploitation correlating with the diversity of the population. The best pareto fronts are those which are providing higher values for both conflicting objects. Figure 4 shows that pareto front values are improved with increasing SINR. Better SINR caused low co channel interference and increased the link capacity.

Figure 5 represents the spectrum efficiency analyses with varying numbers of flows used by IoT nodes for different number of channels. The spectrum increased with increasing flows and vice versa. By using more flows, the spectrum utility is distributed on different flows due to increased spectrum efficiency. Figure 6 represents the energy efficiency versus the number of link flows for different algorithms. Additional power is consumed with growing flows due to reduced energy.

Figure 7 showed that the overall network throughput decreased with increasing spectrum utility because more IoT nodes are accommodated. Moreover, Figure 7 analyzed the convergence performance of different algorithms used in this study. HTSA performs better than TS and SA because it utilized the exploitation feature of TS and research space capability of SA. SA performs relatively better because it explores the search space in comparison with TS. The experiments are conducted on the core i5 system with 6 GB RAM. For N = 30, M = 10, and 500 iterations, the algorithm takes 3.7 ms to converge. The time complexity of the algorithm depends upon the population size and number of iterations performed.

## 5. Conclusions

In this study, a hybrid meta heuristic resource allocation algorithm is proposed to achieve pareto optimality for resource allocation in cognitive radio-based IoT networks. Moreover, fuzzy-based decision making is used to achieve better pareto optimality among conflicting objectives of energy efficiency and spectral efficiency. Furthermore, CR-IoT network performance is analyzed with different network parameters. The proposed algorithm performance is evaluated using simulations, and results are compared with other meta heuristic algorithms. The results showed that proposed HTSA achieve better pareto optimality among conflicting objectives. The inclusion of fuzzy-based decision making in final decision making further improved the optimality of the objectives. The resource allocation for moving nodes in CR-IoT networks is a challenging issue. In future research, we will focus resource allocation on vehicular-based IoT networks [33,34,35].

## Figures and Tables

**Figure 1 sensors-22-00451-f001:**
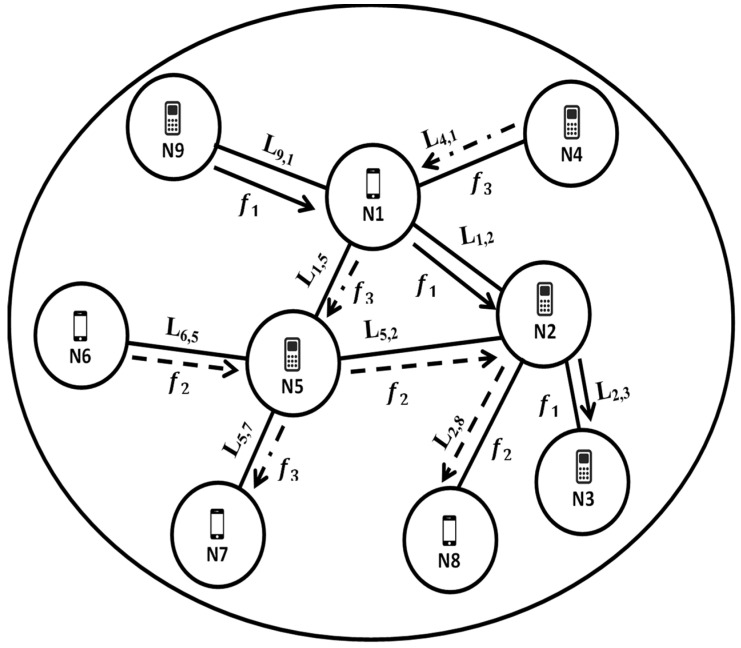
System model.

**Figure 2 sensors-22-00451-f002:**
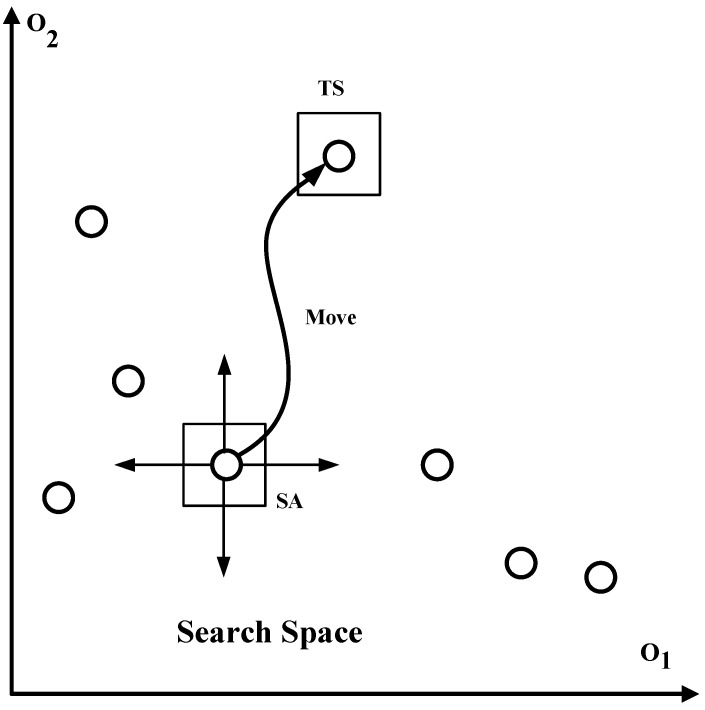
Search process in HTSA.

**Figure 3 sensors-22-00451-f003:**
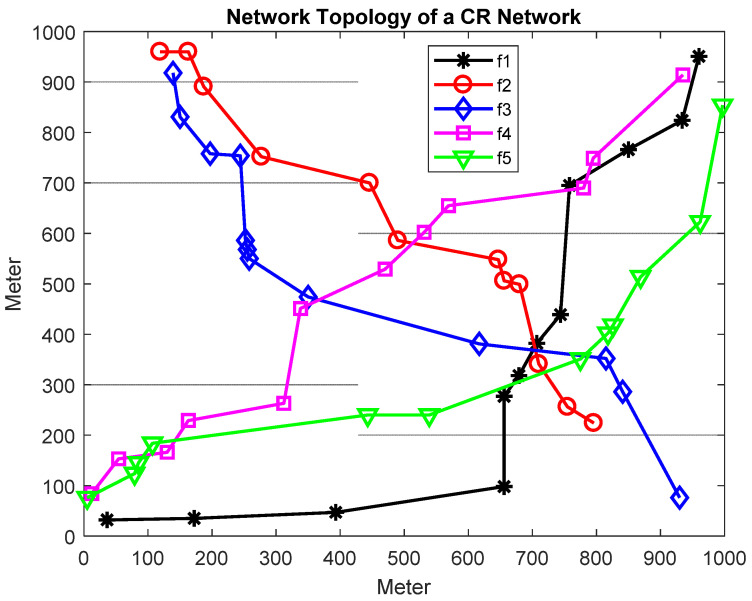
CR-IoT Network Topology.

**Figure 4 sensors-22-00451-f004:**
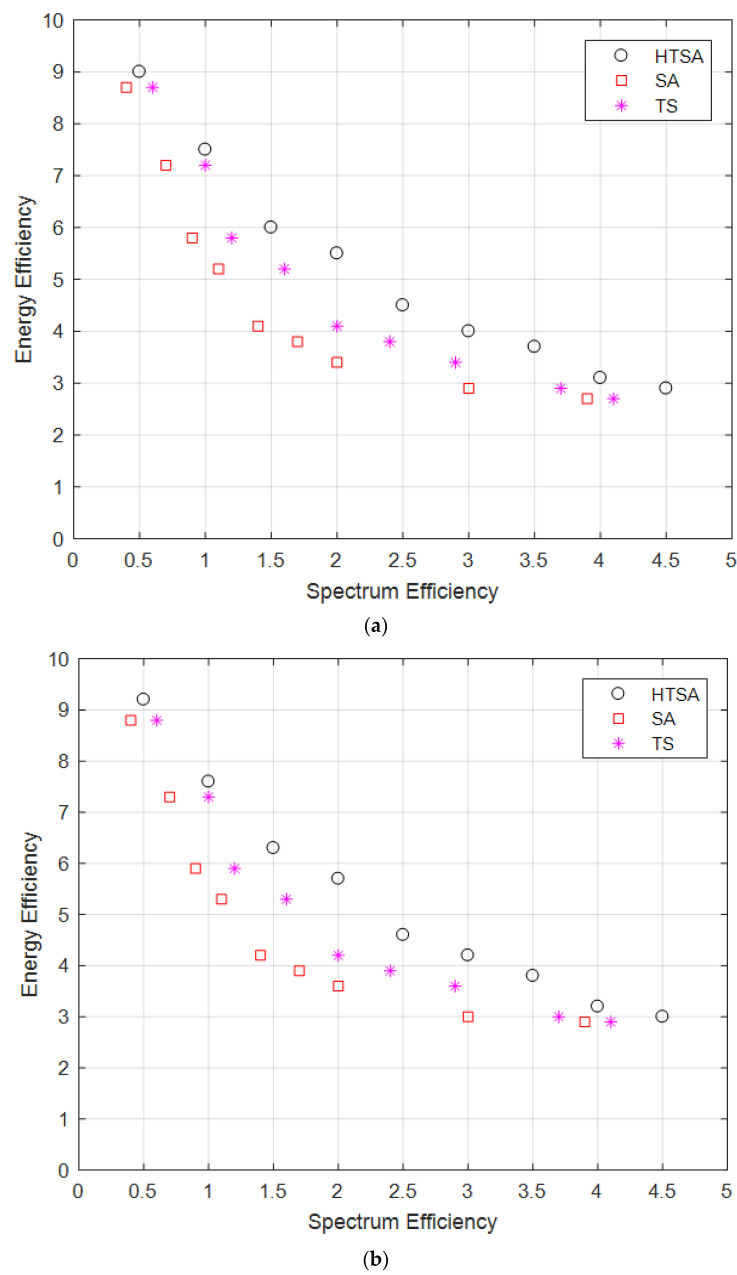
(**a**) γ =4 dB. (**b**) γ =6 dB. (**c**) γ =8 dB. (**d**) γ =10 dB.

**Figure 5 sensors-22-00451-f005:**
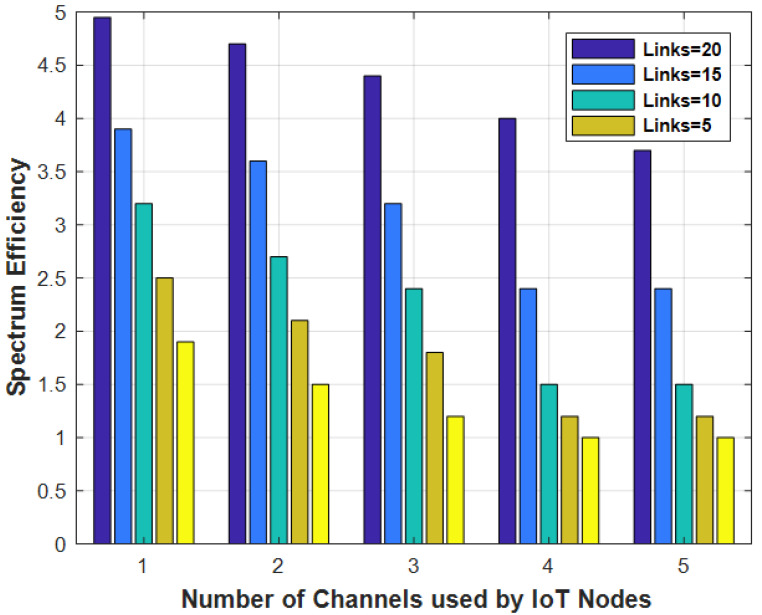
Spectrum efficiency with varying number of channels.

**Figure 6 sensors-22-00451-f006:**
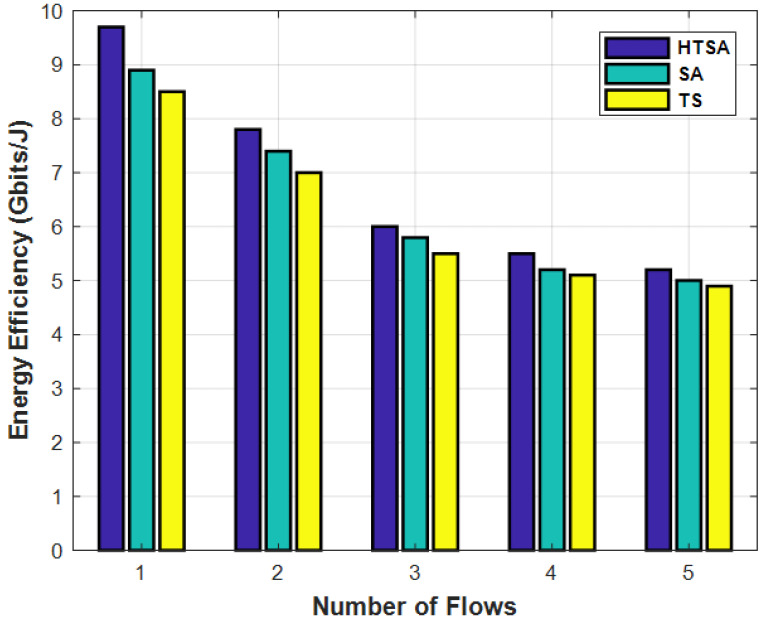
Energy efficiency with varying number of flows.

**Figure 7 sensors-22-00451-f007:**
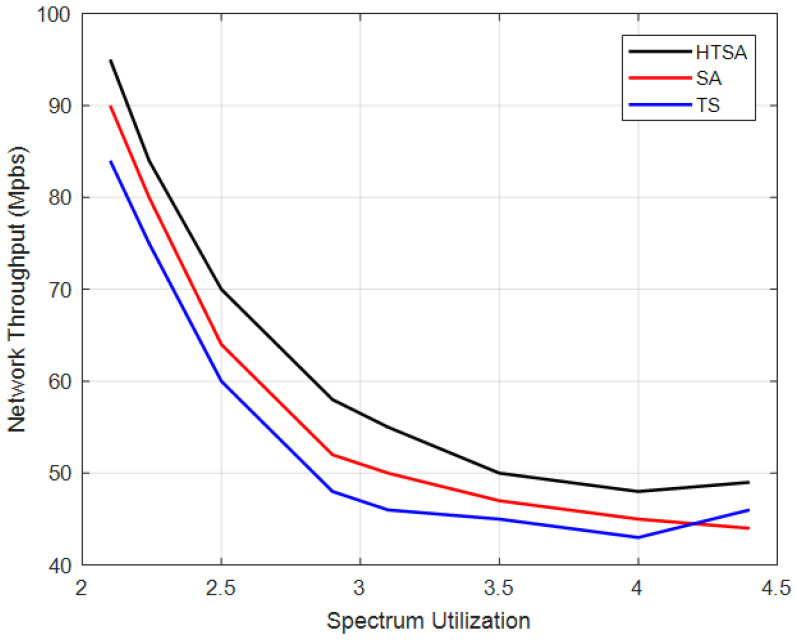
Convergence analyses of heuristic algorithms.

**Table 1 sensors-22-00451-t001:** Initial values of HTSA.

Initial Temperature $\left(T_{i}\right)$	3
Final Temperature $\;\left(T_{f}\right)$	0.000001
Size of Tabu List $\left(T_{L}\right)$	12
Cooling Factor $\left(T_{c}\right)$	0.90
Population Size	50
Number of Iterations	100

**Table 2 sensors-22-00451-t002:** System parameter values.

Number of Nodes (N)	60
Number of Data Flows (F)	5
Number of Links (L)	55
Bandwidth (B)	5 MHz
Number of channels (M)	(5,25)
$P_{min}$	10 mW
$P_{max}$	30 mW
Path Loss Exponent (α)	3
Path loss constant (k)	1

## Data Availability

Not Applicable.

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
