# Peer review of "An Efficient Pareto Optimal Resource Allocation Scheme in Cognitive Radio-Based Internet of Things Networks"

_sensors, 2022, doi:10.3390/s22020451_

Round 1

Reviewer 1 Report

Below, I describe my suggestions that may help improve this article in detail.

  1. Title

The title should reflect its main idea, e.g., a specific approach, method, scenario, novelty aspect, etc. The title of the reviewed paper reflects well the paper's contribution.

In general, most journals recommend using full names instead of abbreviations in titles of articles. However, the IoT abbreviation is very popular and understandable and might be used without abbreviation explication.

  1. Abbreviation

Generally, the authors explain most of the used abbreviations, but there are some missing:

  • Page 1; line 21: NP - first usage Nondeterministic Polynomial - the acronym is popular and understandable but not for all, especially for non-mathematical (optimization issue related) persons,
  • Page 1; line 22: abbreviation for Internet of Things - -IoT is missing.
  • Page 2; line 53: abbreviation for fifth-generation is missing - 5G
  • Page 2, line 57 Multiple Input Multiple Output - abbreviation is missing MIMO.
  1. Content

The introduction provides sufficient background and includes relevant references. The research design is appropriate, and the method is adequately described. The conclusions are supported by the results, which are clearly presented.

In the simulation studies, I missed the evaluation of the computational efficiency of the algorithm, i.e., with the assumed computer equipment, how long it takes to determine the optimal solution.

Page 3, line 101 “The availability of free channels for supporting SUs data transmission depends upon the PUs activity.” – and also SUs activity – not only PUs activity determines the spectrum usage – especially in the contested SUs activity environment like the CRN are.

English language and style are acceptable. The authors shall correct some editorials remarks:

  • Page 3 From line 99 All the parameters/variables (N,M, l(i,j) i^th, j^th, f1, f2, f3) in the shall be ‘italics’ for easier text reading – like in the rest of the paper
  • Page 3, line 116 equations – in the denominator L,(a,b) – is the comma after L is necessary?
  • Page 4 line 145 time t – space is missing
  • Page 4 line 147 – too many spaces ahead N
  • Page 5 line 195 “Figure The  HTS” space is missing between “Figure 2.” and “The HTS…”
  • Generally, in the whole manuscript, double spaces occur – please remove this issue.

Others:

The figures are low-quality/resolution: 1, 2, 4a and 4b and 4c and 4d, 5, 6, 7. So authors shall change them to high-quality ones.

Simulation results

In the simulation studies/results, I’ve missed the evaluation of the computational efficiency of the algorithm, i.e., with the assumed computer equipment, how long it takes to determine the optimal solution.

  1. References

There is a lot of articles related to the topic of this paper. All references are up to date.

Author Response

Response sheet has been attached. thanks

Reviewer 2 Report

Reviewer: In order to handle the combined optimization of conflicting objectives in CR-based IoT networks, this paper proposes a hybrid tabu search-based stimulated algorithm is proposed to achieve Pareto optimality between energy efficiency and spectral efficiency. Moreover, the author employs the fuzzy-based decision to achieve better Pareto optimality and uses different resource allocation parameters to analyze the performance of the proposed algorithm. In general, the paper is well written. However, the reviewer has the following suggestions:

  • What does the temperature mean in the paper, there is no specific equation or symbol to represent it, the authors should explain it.
  • In line 126, the authors claim t as given time instant, what does “t=t*T_c” mean in algorithm 1.
  • In the Results and Discussion section, figure 4 shows that pareto fronts values are improved with increasing SINR. Why don't the authors put the curves of the proposed algorithm with different SINR in a same figure, which will make the analysis more obvious.
  • Some References are missing in the introduction, e.g, “SWIPT Cooperative Spectrum Sharing for 6G-Enabled Cognitive IoT Network, IEEE Internet of Things Journal, 8(20): 15070-15080, 2021.”
  • There are some typo errors in the paper, such as line 24, 116 and 237, please check the paper carefully.

Author Response

(The authors gave the same response as above.)
